# Biofouling Mitigation Approaches during Water Recovery from Fermented Broth via Forward Osmosis

**DOI:** 10.3390/membranes10110307

**Published:** 2020-10-27

**Authors:** Stavros Kalafatakis, Agata Zarebska, Lene Lange, Claus Hélix-Nielsen, Ioannis V. Skiadas, Hariklia N. Gavala

**Affiliations:** 1Technical University of Denmark (DTU), Department of Chemical and Biochemical Engineering, Søltofts Plads 229, 2800 Kgs. Lyngby, Denmark; stkalaf@gmail.com (S.K.); lene.lange2@gmail.com (L.L.); ivsk@kt.dtu.dk (I.V.S.); 2Technical University of Denmark (DTU), Department of Environmental Engineering, Miljøvej 113, 2800 Kgs. Lyngby, Denmark; AGZ@novozymes.com (A.Z.); clhe@env.dtu.dk (C.H.-N.)

**Keywords:** fouling mitigation, biorefinery, forward osmosis, crude glycerol

## Abstract

Forward Osmosis (FO) is a promising technology that can offer sustainable solutions in the biorefinery wastewater and desalination fields, via low energy water recovery. However, microbial biomass and organic matter accumulation on membrane surfaces can hinder the water recovery and potentially lead to total membrane blockage. Biofouling development is a rather complex process and can be affected by several factors such as nutrient availability, chemical composition of the solutions, and hydrodynamic conditions. Therefore, operational parameters like cross-flow velocity and pH of the filtration solution have been proposed as effective biofouling mitigation strategies. Nevertheless, most of the studies have been conducted with the use of rather simple solutions. As a result, biofouling mitigation practices based on such studies might not be as effective when applying complex industrial mixtures. In the present study, the effect of cross-flow velocity, pH, and cell concentration of the feed solution was investigated, with the use of complex solutions during FO separation. Specifically, fermentation effluent and crude glycerol were used as a feed and draw solution, respectively, with the purpose of recirculating water by using FO alone. The effect of the abovementioned parameters on (i) ATP accumulation, (ii) organic foulant deposition, (iii) total water recovery, (iv) reverse glycerol flux, and (v) process butanol rejection has been studied. The main findings of the present study suggest that significant reduction of biofouling can be achieved as a combined effect of high-cross flow velocity and low feed solution pH. Furthermore, cell removal from the feed solution prior filtration may further assist the reduction of membrane blockage. These results may shed light on the challenging, but promising field of FO process dealing with complex industrial solutions.

## 1. Introduction

Forward osmosis (FO) is an emerging technology that has attracted great attention during the past decade. This technology is using semipermeable membranes to extract water from various aqueous solutions. The main driving force is the osmotic pressure difference between the feed and draw solution [1]. FO offers several advantages over the energy demanding hydraulic pressure-driven processes, e.g., nanofiltration (NF) and reverse osmosis (RO). Among others, FO is characterized by high solute rejection, low energy requirement, reduced fouling and simple cleaning [1]. Therefore, FO has a potential for becoming a sustainable and economically feasible process for large scale water recovery applications. 

FO application has been mainly investigated in the fields of wastewater and seawater desalination [2,3,4]. Nevertheless, later studies have investigated the potential of using FO in 2nd generation biorefinery [5]. This is appealing as vast amount of water is required for diluting 2nd generation feedstocks mainly due to substrate and product inhibition [6]. This has an adverse effect on the process sustainability because of high water consumption and the great amount of energy requirements for product recovery. Therefore, the use of FO can pose great benefits. 

Crude glycerol is a commercially interesting 2nd generation biorefinery feedstock. The rapid growth of biodiesel industry has created a large surplus of crude glycerol corresponding to one-tenth of the production volume during of the biodiesel industry [7]. Glycerol can be converted via anaerobic fermentation into various products like butanol, ethanol, and 1,3-propanediol [8]. For example, *Clostridium pasteurianum* has been reported to produce up to 17 g/L of butanol during glycerol fermentation [9]. Nevertheless, this product concentration is rather low; thus, FO can be used for product up-concentration and consecutive reduction of the recovery costs. Even though FO shows a great potential to recover water from fermentation effluents, several challenges have been reported regarding the operational conditions and the water recovery efficiency during application of FO with the use of fermentation effluents [10].

Specifically, fouling is inevitable during filtration processes and is caused by deposition of particles (inorganics, colloids, and organics), cells and cell debris on the membrane surfaces, thus hindering water permeability [11,12,13]. Furthermore, solutes are accumulated in the vicinity of the membrane, creating an interfacial water layer with an osmolality different from the bulk solution [1]. This effect is called concentration polarization (CP) and can be divided into internal (ICP) and external concentration polarization (ECP) corresponding to solute accumulation or depletion inside or outside the membrane material [1].

Biofouling is probably the most detrimental type of fouling found in membrane processes and it can account for up to 40% of total fouling during reverse osmosis filtration [14]. Biofouling is caused by attachment of cells, cell debris, extracellular polymeric substances (EPS), proteins, and polysaccharides on the membrane surface that leads to a water flux decrease [14,15]. Further, fouling layers have been reported to improve the solute rejection by blocking their diffusion, consumption of organic foulants, or even changing the membrane properties [16,17]. Generally, it has been reported that biofouling amelioration is much easier in FO processes, in comparison to RO, as the fouling layer is formed as a loose attachment of the foulants on the membrane surface and not as a pressure-induced compaction layer [15].

Various biofouling mitigation approaches have been investigated to improve the performance of FO such as FO membrane modification, optimized operation and feed pretreatment. For instance, to limit the growth of bacteria on FO membrane, an increase of feed cross-flow velocity has been found to have a great impact on FO performance by reducing bacterial approach to membrane surface and EPS attachment [18,19]. As reported by Lee et al. [20], a high feed flow rate has been found to alleviate fouling in FO system, whereas a high draw flow rate was found to enhance fouling growth, due to a reverse salt flux. Furthermore, nutrient load control, phosphate limitation, and selection of appropriate draw solutions have been proposed as biofouling alleviation strategies [11,21]. Additionally, changing pH can affect attachment of foulants such as octanoic acid and proteins [12,22] and also alter the membrane charge [23]. Furthermore, the effect of membrane orientation in CP has been extensively studied with a general consensus reporting that an active layer-feed solution (AL-FS) orientation is the optimal configuration [24]. Other fouling mitigation approaches include coating of active membrane side and modification of spacer geometry [25] and finally physical/chemical cleaning [14].

This study is focused on identifying the crucial operation parameters for biofouling amelioration while applying fermentation effluent and crude glycerol as feed and draw solutions, respectively, in FO processes using biomimetic aquaporin membranes. The initial microbial biomass concentration, feed solution pH and cross-flow velocity were studied for their effect on biofouling and the corresponding water recoveries. Finally, the correlation of biofouling and membrane blockage with reverse glycerol flux and process butanol rejection was also investigated.

## 2. Materials and Methods

### 2.1. Feedstocks and Treatment

#### 2.1.1. Crude Glycerol

Crude glycerol was kindly provided by Daka ecoMotion (Løsning, Denmark) and is a biodiesel by-product. Detailed composition of this feedstock is provided elsewhere [26].

#### 2.1.2. Organic Solvent Treatment of Crude Glycerol

Anhydrous n-Hexane 95% (Sigma-Aldrich, St. Louis, MI, USA) was used for extraction of unsaturated fatty acids from crude glycerol. A protocol described elsewhere [27] was applied with minor modifications. In short, potassium hydroxide pellets (KOH, ACS reagent grade, ≥85% purity, Sigma-Aldrich, St. Louis, MI, USA) were dissolved in a crude glycerol solution until pH 6.0 was reached. Following, the solution was centrifuged at 4400 rpm for 15 min and the supernatant was collected and underwent hexane washing/treatment. Equal amounts of n-hexane (anhydrous, purity 95%, Sigma-Aldrich, St. Louis, MI, USA) and crude glycerol were placed in 500 mL glass bottles and agitated for 3 h at 200 rpm. Then, the liquid was placed in a 2 L separation funnel (Figure 1) and left to separate for 1.5 h. Finally, the lower phase was collected and subjected to a second washing–separating cycle. At the end of the process, treated crude glycerol containers was kept open in fume cupboard in order to evaporate any residual hexane. The purified glycerol was used for fermentation with *C. pasteurianum* and for FO as draw solution.

#### 2.1.3. Total Suspended Solids

Total suspended solids were measured with Whatman^TM^ glass microfiber filters, 0.7 µm particle retention (GE Healthcare Europe GmbH, Bröndby, Denmark) as described in the “Standard methods for examination of water and wastewater” [28].

### 2.2. Strain and Fermentation Conditions

The strain *Clostidium pasteurianum*, DSM 525 (DSMZ, Göttingen, Germany) was used for crude glycerol fermentations. A medium described elsewhere [29] was used during the fermentations with reduced FeSO_4_ (7.5 mg L^−1^) concentration [30]. The fermentation conditions, fermenter and control unit are described elsewhere [10]. In short, 3 L glass bioreactors (Applikon^®^, Delft, The Netherlands) with and ez-control unit (Applikon^®^, Delft, the Netherlands) were used for conducting the fermentations. Fermentations took place at 37 °C, 200 rpm agitation and pH 6.0. 

### 2.3. Forward Osmosis Membranes and Operation

#### 2.3.1. Membrane Coupons and Flow Cells

Aquaporin-based biomimetic thin film composite flat-sheet membranes were used during the experiments (provided by Aquaporin A/S, Kgs. Lyngby, Denmark). The membrane specifications are described elsewhere [31]. The membrane coupons, yielding an active filtration area of 33.15 cm^2^, were placed in a FO chamber (85 mm long, 39 mm wide and 2.3 mm channel height, filtration area 33.15 cm^2^), and spacers (30 mil/762 µm thickness, Alfa Laval, Lund, Sweden) were applied on both sides of the membrane [10].

#### 2.3.2. Operational Parameters

The general operational parameters are described elsewhere [10]. In short, membranes were pretreated for 30 min in Milli-Q water, then placed in the filtration flow cells, and subsequently the corresponding feed and draw solutions were loaded in the system. Feed solutions were always placed towards the active membrane layer and FO operation was performed in counter-current flow. Both feed and draw solutions were agitated at 180 rpm with a magnetic stirrer and the feed solution was kept at 37 °C with a silicone heating mat (Lund & Sørensen A/S, Vejle, Denmark).

#### 2.3.3. Forward Osmosis Baseline Experiments

Prior to each FO experiment using fermentation broth as feed solution for FO, baseline experiments were conducted for 20 h. Two liters of Milli-q H_2_O and 1 M NaCl solution (purity ≥ 99.5, Sigma-Aldrich, St. Louis, MI, USA) was used as FS and DS, respectively. Both solutions were continuously mixed with a magnetic stirrer (180 rpm) and the FS temperature was maintained at 37 °C with a heating mat (Lund & Sørensen A/S, Vejle, Denmark). Cross-flow velocity of 4.7 cm s^−1^ applied for the membrane coupons that were subsequently used for the experimental set-ups 1 and 2; or 19.4 cm s^−1^ for the coupons to be used for set-ups 3, 4, and 5. Both conductivity and mass of the FS were monitored on-line (5 min intervals). The cross-flow velocity of the FS was monitored through a ZYIA (10-100 LPH) liquid flowmeter/regulator (Roykon, Fredericia, Denmark). After the 20 h baseline experiments, the performance of each membrane was assessed. The minimum acceptable *J_S_/J_W_* was empirically set to 0.4 g L^−1^; if the membrane did not fulfill that requirement it was discarded and a new coupon was used for a new baseline run. Preliminary experiments have shown that for a consistent performance, the minimum baseline reverse salt flux should not exceed 0.4 g.L^−1^. Moreover, and according to Blandin et al. [32], a J_S_/J_W_ of 0.4 g.L^−1^ was within the performance range of other commercially available FO membranes. If an acceptable *J_S_/J_W_* was reached then the system was washed twice with 2 L of Milli-q H_2_O (conductivity at DS < 30 µS cm^−1^, FS < 1 µS cm^−1^) and consecutively the membrane experiments were performed using fermentation broth and hexane treated crude glycerol as FS and DS, respectively.

### 2.4. Biofouling and Other Organic Fouling Experiments

#### 2.4.1. Experimental Parameters

A number of experimental parameters have been identified for the analysis: (i) initial FS composition (cell-free or with cells), (ii) cross-flow velocity (low: 4.7 cm s^−1^, high: 19.4 cm s^−1^), and (iii) pH of the FS (close to neutral, 4.0 and 8.4). 1 M HCl and 4 M KOH solutions were used for pH adjustments to 4.0 and 8.4, respectively. Furthermore, the effects of one or a combination of parameters were studied. A graphical representation of the experimental design and the relevant effects under investigation is shown in Figure 2. All experimental setups were performed in two individual experiments using always new membrane coupons. The initial volume of the corresponding feed solutions was 2 L and hexane treated crude glycerol (300 mL) was always used as feed solution. 

#### 2.4.2. ATP Analysis: Total and Free-ATP Quantification on the Membrane Surface

Immediately after the filtration and prior to the ATP analysis, the membrane was removed, cut in half and stored at −80 °C as showed in Figure 3. A protocol described before was applied for the ATP extraction [33]. Before the analysis, the membranes were removed from the freezer and two 3 × 3 cm samples were cut (i) one from the inlet and (ii) one from the membrane outlet (see Figure 3). 

Subsequently, the samples were transferred to 50 mL falcon tubes and 50 mL of ATP-free water was added. The samples were sonicated for 6 min, to disrupt possible biofilm, vortexed three times for 10 s and then the cell suspension was stored at −80 °C prior the ATP analysis (6 days later). The ATP analysis was performed according to a protocol described by Vang and colleagues [34] and separate total and free-ATP calibration curves were prepared. The ATP samples were measured in duplicate. Standard deviation between the technical duplicates was below 10%. Two samples with autoclaved tap water were used as separate control during Total and Free-ATP quantification.

#### 2.4.3. Calculations—Water Flux, Reverse Glycerol Flux, and Process Butanol Rejection

The weight of the feed solution was monitored every 5 min (FCB 24K2 Kern balance, KERN & SOHN GmbH, Balingen, Germany) and the values were used for calculating the respective water fluxes. 

Liquid samples for the calculation of reverse glycerol flux and the process butanol rejection were taken from the feed solution every 30 min for the first 6 h and at the end of the filtration (22 h).

Reverse glycerol flux was calculated from the feed solution mass balance according to Equation (1) [11,35]: (1)$$J_{g}=\frac{C_{\mathrm{Ft}}*\left(V_{\mathrm{Ft}0}-\Delta V\right)-C_{\mathrm{Ft}0}{*V}_{\mathrm{Ft}0}}{A*\Delta t}\;\mathrm{in}\;\mathrm{mol}\;m^{-2}\;s^{-1}$$
where *J_g_* is the reverse glycerol flux, *C_Ft0_* corresponds to the initial glycerol concentration of the feed solution, and C_Ft_ to the concentration of glycerol at a given sampling point. Additionally, V_Ft0_ is the initial volume of the feed solution (always 2 L), ΔV is the volume decline of the feed solution, and Δt is the time difference between the corresponding measurements. 

The butanol rejection was calculated according to Equation (2):(2)$$R_{BuOH}\left(\%\right)=\left(1-\frac{C_{\mathrm{Ft}0}*V_{\mathrm{Ft}0}-C_{\mathrm{Ft}}\left(V_{\mathrm{Ft}0}-{\Delta V}_{F}\right)}{C_{\mathrm{Ft}0}*V_{\mathrm{Ft}0}}\right)*100$$
where V_Ft0_ is the original feed filtration volume (2 L); ΔV is the volume decline; and C_Ft0_ and C_Ft_ are the concentrations of butanol at the beginning and at the corresponding sampling point, respectively.

### 2.5. Analytical Methods

#### 2.5.1. High Performance Liquid Chromatography (HPLC)

Organic solutes (butanol and glycerol) were quantified by HPLC through an Aminex HPX-87H column (Bio-Rad, Hercules, CA, USA). HPLC was equipped with a refractive index detector and sulfuric acid 12 mM (Sigma-Aldrich, St. Louis, MI, USA) was used as eluent at a constant flowrate (0.6 mL min^−1^).

#### 2.5.2. ATR-FTIR

For the identification of functional group contributing to membrane fouling, Fourier transform infrared spectroscopy (FT-IR; Spectrum 100 Spectrophotometer, PerkinElmer, MA, USA) was used. The membrane preparation and the spectra measuring conditions used are described elsewhere [10]. In short, several measurements were taken along both membrane sides within a range of 4000 to 650 cm^−1^ with 4 cm^−1^ resolution.

#### 2.5.3. Scanning Electron Microscopy (SEM)

Fouled membrane samples for examination by SEM were retrieved from the module at the end of operation and stored at 4 °C prior to analysis. The membrane surface structure and morphology was examined using a scanning electron microscope (SEM) (FEI Quanta 200 ESEMTM, FEG, Hillsboro, OR, USA). Samples were dried at room temperature and gold coated as previously described [36]. The coated membrane specimens were studied at an accelerating voltage of 10 kV and a spot size of 3.0 nm at a working distance of 10 mm.

## 3. Results and Discussion

### 3.1. Baseline Experiments

Baseline experiments were conducted to acquire representative performance characterization, in terms of reverse salt flux and water flux, for each membrane coupon to be used for the experiments. The results from the average *J_Waver_* and *J_Saver_* during the 20 h of the baseline experiments are presented in Table 1*. J_S_/J_W_* represents the specific reverse solute flux (SRSF), in this case 1 M NaCl solution.

As shown in Table 1, the majority of the membrane coupons performed consistently regarding the *J_S_/J_W_* ratio. At a review of 2016 [32], the performance characteristics of various flat-sheet membranes and reported *J_S_/J_W_* ratios between 0.1 and 1.3 g L^−^^1^. As shown in Table 1, the obtained results using Aquaporin Inside^TM^ membranes exhibited high performance. The latter can be attributed to the biomimetic nature of the membranes that facilitate water transfer through aquaporin channels. Those channels are embedded on the membranes of many organisms and especially AqpZ proteins facilitates pure water transfer [37]. Interestingly, only a minor increase of *J_Saver_* was detected in correlation to the increasing CFV. Finally, for all membranes *J_S_/J_W_* < 0.36 g L^−1^ and performed consistently in terms of reverse glycerol flux and process butanol flux in the corresponding following experiments, described in Section 3.3.

### 3.2. Biofouling

#### 3.2.1. Water Recovery and ATP Accumulation

Adenosine triphosphate (ATP) analysis is generally applied for assessing wastewater and drinking water quality [34] and ATP is the main cellular energy storage molecule [38]. Thus, ATP concentration (typically measured as ng cm^−2^) can generally be used for assessing cellular activity and viability and the concentration of free ATP is generally considered as cell decay indicator [39]. In the context of this study, Total and Free-ATP analysis has been employed to correlate biofouling on the membrane surface with water recovery. Although the correlation of ATP concentration with cell concentration is not always linear and it can be affected by the physiological stage of the cells and cell species [38], ATP analysis remains a useful indicator for cellular accumulation on membrane surfaces [40,41,42]. The Total-ATP detected on the membrane after 22 h of FO operation and the respective water recoveries are discussed in this section. Water recoveries are expressed in L m^−2^ (water recovered per m^−2^ of membrane area) for the total experimental duration, whereas ATP represents the average Total-ATP cm^−2^ quantified on the corresponding membrane sample areas as indicated in Figure 3. The different setups studied and the possible effects are shown in Figure 2 (Section 2.4.1).

Bucs and colleagues [40] showed that biomass accumulation had an important effect on permeate flux decline during consecutive filtration cycles. Hereby we report, as shown in Figure 4, that in most of the cases water recoveries are inversely proportional to the Total-ATP detected on the membranes, namely, the higher the Total-ATP concentration on the membrane, the lower the water recovery is, with the only exception of set-up 4 (see Section 3.2.2, Paragraph 2). 

One may observe that there is a variation of ATP concentration between the different setups and also when identical operational conditions have been applied. For example, the Total-ATP count between the two experiments in set-up 1 is the most noticeable example: 57 ng cm^−2^ and 18 ng cm^−2^ of Total-ATP for set-ups 1A and 1B, respectively. This can be explained by the mechanism of biofouling development. Namely, biofouling formation can be identified in three stages: (1) initial attachment of proteins and macromolecules; (2) attachment and initial growth of microorganisms; and (3) further growth and build-up of biofouling layer by the production of extracellular polymeric substances (EPS), proteins, and polysaccharides [15]. Many parameters can affect the settlement and biofilm prolongation. For example, organic and inorganic compounds are transferred to the vicinity of the membrane due to Brownian motion and cells can be actively moved close to the membrane surface by chemotaxis [43]. Other foulant–membrane interactions include membrane hydrophobicity/hydrophilicity, membrane charge, bacterial size, shape and contact angle, permeation drags, electrostatic, and van der Waals interactions [44]. All in all, the results of the current study together with the literature mentioned above provides evidence that fouling accumulation is challenging to predict when complex media are used, such as fermentation broth as FS, due to complex interactions that take place during filtration.

Therefore, for the 22 h experiments applied here the probability of achieving similar Total-ATP concentrations is rather low, even when identical operational conditions are applied (e.g., 1A and 1B). However, it is expected to have similar ATP concentration during long-term filtration when the biofilm is fully developed. For osmotic membrane bioreactors (OMBRs), it has been shown that 8 days were required for the biofilms to be fully developed. In this case biofilm formation led to a water flux decline, though after the biofilm settlement the water flux remained stable [45].

Additionally, the process conditions can also affect the cell accumulation. Thus, when higher cross-flow velocities were applied, the reproducibility of the water recoveries and Total-ATP concentration was improved. By the results of this study, it is postulated that at low CFV the initial deposition of foulants is more unpredictable than at high CFV, probably due to the lower linear velocity and shear force taking place at low cross-flow velocity. As mentioned above, initial fouling is followed by rapid cellular accumulation. Thus, it is hypothesized that during the ATP accumulation measured at 22 h, fouling is present at different stages. Hence it is expected that in longer term experiments the difference in ATP accumulation on the membrane surface, when same operation conditions are applied, would be much lower. Finally, the physiological state of the microbes might also have an effect on cells attachment on the membrane surface, see also Section 3.2.3.

Last, regarding the FS pH 4.0 (set-up 4), even though the Total-ATP accumulation on the membranes was found low, in comparison to experiments conducted at neutral or alkaline conditions (Figure 4), a water recovery equivalent to intermediate Total-ATP accumulation was obtained. As revealed by SEM analysis (Appendix A) the active membrane side incorporated with the aquaporin channels is not fully covered by a cell layer as it is in case of setups 1–3 and 5 (Appendix A). However, in some areas, the membrane may be covered by EPS and other organic/inorganic colloids.

In order to have a better overview of the fouling compounds accumulated on the membrane surface, ATR-FTIR was used to investigate changes in surface chemical compositions of Aquaporin Inside^TM^ membranes after FO operation. Firstly, ATR-FTIR spectra of clean membrane have been collected as controls (Appendix A) and the main functional groups identified as belonging to polyamide (PA) and polyethersulfone (PES) (Appendix A) in agreement with previous studies [46,47,48,49]. The fouled active side of the membrane layer exhibited bands at 3279, 2927, 1634, 1536, and 1043 cm^−1^, which were absent in spectra of the clean membrane. The bands at 1634, 1536 cm^−1^ are characteristic for proteins [50], whereas the bands at 3279, 2927, and 1043 cm^−1^ are characteristic for –CH and –OH groups, commonly attributed to polysaccharides [51]. This is consistent with formation of a proteinaceous saccharine layer and the results are in good agreement with previous studies using similar feed solutions [10]. Furthermore, both ATP and SEM analysis confirm the presence of biofouling on the active side of membrane. It can be also noticed from Appendix A that the characteristic intensities of the clean membrane are significantly decreased on the fouled membranes, due to mounting of deposits on the membrane surface (Appendix A). Interestingly, for setup 4, the protein signal was significantly reduced, whereas the signal for polysaccharides was similar to the fouled membranes. This suggests that the main foulant at low pH was associated with polysaccharide accumulation on the active membrane layer. Another possible scenario explaining the obtained low water recovery for set-up 4, could be the increase of membrane hydrophobicity at low pH as reported by Hurwitz and colleagues [52], thus hindering water transport across the membrane [52].

Finally, it is noteworthy that ATR-FTIR did not reveal any glycerol deposition on the support membrane layer (Appendix A) which is supported by SEM study (Appendix A). Thus, these results further confirm previous studies [10], where it is reported that that crude glycerol works well as a draw solution in FO process.

#### 3.2.2. Effect of Initial Cell Concentration, Cross-Flow Velocity and pH in the Cellular Accumulation

The highest Total-ATP concentration was detected when low cross-flow velocity and high initial cell concentration were applied. Subramani and Hoek reported for RO membranes that when small microorganisms are present in the filtration solutions, electrostatic repulsion is the prevailing mechanism of initial cellular deposition [44]. Thus, when low cross-flow velocity is applied, the electrostatic repulsion is the most dominant force for the initial biofouling attachment. However, when cross-flow velocity is increased, hydrodynamic forces prevail over electrostatic repulsive forces, which are responsible for the initial biofilm formation. Furthermore, increased flow rate can facilitate the partial biofouling alleviation during FO operation, due to greater shear forces that can rinse the membrane from foulants [44]. This can be seen in Figure 4, where a reduced amount of Total-ATP was detected for set-up 3 (with a flow rate of 19.4 cm s^−1^) when comparing to set-up 2 (with a flow rate of 4.7 cm s^−1^). This is also evident from the normalized Total-ATP/TSS ratio shown in Figure 5. This is in agreement with the studies of Boo and colleagues [53] that indicated membrane fouling amelioration in high cross-flow velocities (32.1 cm s^−1^). The latter can be attributed to the detachment of biofilm layer and less foulants’ accumulation leading to formation of thinner and looser fouling layer [15]. In case of fouling in spiral wound FO membrane induced by secondary wastewater effluent, a flow rate of 5.54 cm s^−1^ was found to be sufficiently high to mitigate fouling [20], which was not the case in our study. This indicates an importance of finding appropriate hydrodynamic operating conditions, which are case specific. On the contrary, Bucs and colleagues [40] reported increased biofouling at higher cross-flow velocities (16.3 and 24.5 cm s^−1^), possibly due to increased nutrient transport rate to the deposited film resulting in a higher biomass growth.

Regarding the effect of FS’s pH on the cell deposition on the membrane fouling, the best results were obtained under acidic conditions (pH 4.0). This is an interesting finding, since the opposite trend would have been expected due to electrostatic repulsion forces between the cells and the membrane. For example, the zeta potential of two bacterial species (suspended in a nutrient-free medium) has been measured to be −33 and −41 mV [44]. Thus, one would expect a stronger tendency for cells to accumulate on the membrane, due to decreased electrostatic repulsion at low pH, since the Aquaporin Inside^TM^ membrane is also less negatively charged [23]. However, in this study we found low cellular deposition at pH 4.0. This is clear evidence that in complex feed solution systems, like fermentation broths, other interactions than purely electrostatic might become prevailing including potential changes of the cellular membrane physiological properties. It is known that the hydrophobicity and membrane charge of the cells can vary depending on the physiological cell stage and the nutrient composition [54,55]. In the present study a possible explanation is that *C. pasterianum* cells, with an optimal pH growth between 4.5 and 7.5 [29], might be inactivated at pH 4.0 thus presenting reduced charge on the membrane surface. Moreover, other interactions between the proteins, polysaccharides, and ions present in the fermented effluent might have an important effect on the deposition/repulsion of the cells. For example, it has been reported that in low pH, calcium cations tend to compete with other compounds for adhesion sites on the membrane surface, thus reducing their probability to deposit [12]. Finally, the presence of the extracellular polymeric substance might promote the development of a conditioning film on the membrane surface [56]. The results presented in Figure 4, are supported by SEM analysis showing only few cells attached on the membrane surface for set-up 4 (Appendix A). On the contrary, under alkaline conditions (pH 8.4), no significant reduction of Total-ATP was observed (set-up 5, Figure 4). However, in this particular case there was a pH drop during the filtration (Table 2). This might be attributed to (a) transfer of anions from the feed to draw solution, (b) back-diffusion of crude glycerol, (c) bacterial growth metabolites (e.g., organic acids) that are produced in the FS during the filtration leading to pH decrease, or (d) a combination of the above phenomena.

#### 3.2.3. Effect of Cellular Activity on Biofouling

The initial biomass concentration had a significant effect on the biofouling formation and consequently on water recovery, with the set-up without cells in the feed solution (setup 1) exhibiting higher water recovery, even at lower CFV and neutral pH. This is a strong indication that a process step targeting cells removal could be advantageous.

As discussed in Section 3.2.1, when comparing set-up 1 (no cells in the FS) with set-up 2 (with cells in the FS), there is an increase of Total-ATP in setup 2 (Figure 4). However, it can be hypothesized that active cells tend to attach more on the membrane. This is supported by the intracellular ATP/Total-ATP (Table 3). As mentioned above, high extracellular ATP indicates that the microbial populations are entering the cell decay stage [39]. Therefore, high intracellular/Total-ATP ratio can indicate highly active cells.

As shown in Table 3, the intracellular/Total-ATP ratio in set-up 1 was between 0.79 and 0.95, which is much higher than in any other conditions investigated here. This can indicate that most of the cells present on the membrane were active. In the rest of the setups (apart from set-up 4, where ATP was very low) the ratio mentioned above was always below 0.47. Furthermore, as shown in Figure 5 when the Total-ATP is normalized by the total TSS (in g per L^−1^) present in the feed solution at the end of the filtration the tendency of the cells to attach on the membrane is the highest (setup 1). The latter is another indication that cells have higher tendency to induce biofouling, when they are active. This result is supported by the SEM analysis. In Appendix A is shown that the biofilm layer was very compact in set-up 1 and almost no cells could be identified, since other material were tightly attached to the fouling layer. This can be attributed to high EPS, proteins and polysaccharides incorporation in the biofilm, caused by high cellular activity. Kwan and colleagues [15] reported that in the FO process, biofouling is structured in three layers: dead cells are accumulated close to the membrane surface, covered by an active cell layer that is enclosed in a thick polysaccharide layer [15]. Therefore, the majority of the compounds in the external layer are polysaccharides and EPS [45,57]. Second, the higher water flux has probably lead to more compact fouling [58]. Last, the SEM analysis suggests that less active cells tend to produce less EPS. As shown in Appendix A even though setup 2 and 3 had very high cell density, still biofilm was not as compact as for set-up 1 (Appendix A). Furthermore based on the SEM analysis, no evidence of salt crystallization was found (Appendix A).

#### 3.2.4. ATP Accumulation in the Membrane Inlet and Outlet

As shown in Figure 6, there was a general tendency for higher Total-ATP accumulation on the membrane inlet for set-up 1, 2, and 3. This observation is in agreement with reports of higher biomass accumulation in the lead of membrane modules, due to higher slope of the pressure drop profile at the beginning of membrane module, which induces the compactness of foulants at the inlet [42]. Another study attributes the ATP concentration decrease to the decline of nutrient availability along the membrane surface, therefore the reduction of biofilm formation [41]. However, for setup 5 similar Total-ATP concentrations have been detected at the membrane inlet and outlet. It is difficult to draw conclusions though the reason behind this phenomena, as the FS’s pH dropped during the experiment, as shown in Table 2. 

### 3.3. Specific Glycerol Reverse Flux and Process Butanol Rejection

The reverse glycerol flux results are well in accordance with the membrane autopsy results. Figure 7 shows that the specific glycerol reverse flux rates (*J_g_/J_W_*) for set-up 1, 2, and 3 are much lower than for set-up 4, where low ATP and limited number of cells were detected by SEM analysis (Appendix A). In setups 1–3 there was high biofilm formation resulting in production of an additional barrier for glycerol back-diffusion. Thus, due to biofilm formation on the membrane surface, glycerol back-diffusion is prevented. Furthermore glycerol might be also consumed by the microorganisms attached on the membrane surface [16]. Finally, glycerol can be utilized by the bacteria incorporated in biofilm, hence enhancing the biofilm formation with simultaneous reduction of glycerol concentration in the bulk solution.

As shown in Figure 7, glycerol reverse flux is stable during the first 6 hours of the experiments irrespective of the operational parameters. However, later and when the biofilm is formed and bacteria start to catabolize glycerol, the specific reverse glycerol flux is decreased towards the end of the experiment for set-ups 1, 2, and 3. On the other hand, when the biofilm layer is absent and there is limited cellular growth on the membrane surface (set-up 4) there is no barrier to retain the glycerol; thus possibly more glycerol is diffusing across the membrane. For the alkaline pH the results were inconclusive. All in all, specific glycerol reverse flux was much lower in all the studied scenarios than the one obtained in previous experiments [10]. This in combination with acceptable process butanol rejection (73–88% without a clear pattern across the setups), is an encouraging development towards the optimization and application of biomimetic Aquaporin Inside^TM^ membranes for water recovery purposes in 2nd generation biorefinery.

In summary, the experimental results obtained in this study provide important insights regarding cellular and extracellular material interactions with Aquaporin Inside^TM^ membranes. Specifically, substantial water recovery, low specific reverse glycerol flux and high process butanol rejection are possible to achieve, when optimal FO operation conditions are applied, namely: high cross-flow velocity, pH 4.0 and separation of cells prior to the FO process step.

## 4. Conclusions

This study provides important new findings regarding biofouling and possible mitigation strategies when 2nd generation biorefinery effluents and feedstocks are applied as feed and draw solutions for water recovery via FO filtration:Water recovery can be directly linked with Total-ATP accumulation on the membrane. In the case of low pH other factors can affect the membrane blockage and water recovery, e.g., electrostatic repulsions/attractions between the membrane, the cells, and extracellular material in the feed solution.ATP analysis revealed that cells on the exponential growth face are more prone to attach on the membrane surface resulting in higher degree of biofouling.As revealed by ATR-FTIR analysis of the support layer of the membrane did not suffer from severe fouling (ATR-FTIR), or biofouling as demonstrated by the SEM study.Total-ATP was higher at the inlet than at the membrane outlet. This knowledge can be used when considering switching flow direction as a biofouling mitigation approach.Low pH has the greatest impact on cellular accumulation on the FO membrane surface. However, the highest specific glycerol reverse flux was obtained. This can be explained by the lack of biofilm development on the membrane surface and lower glycerol consumption by microorganisms.Process butanol rejection was found to be between 73 and 88% during 22 h filtration experiments.

In overall, a cell removal step from the feed solution prior filtration, low pH and higher CFV showed high potential for alleviating FO biofouling.

## Figures and Tables

**Figure 1 membranes-10-00307-f001:**
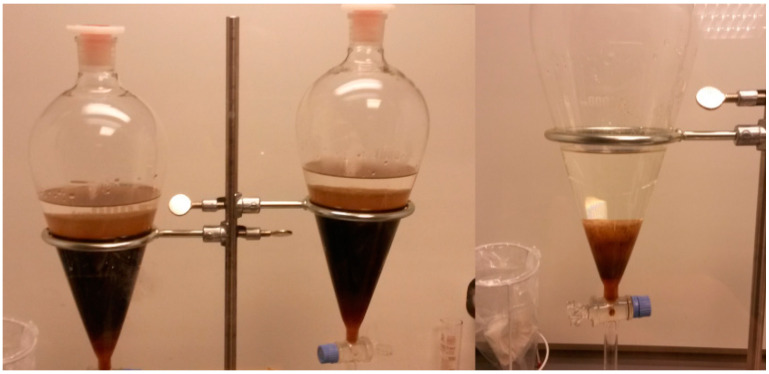
Hexane extraction step. Three zones are formed: Top: hexane; intermediate: hexane and unsaturated fat; bottom: purified crude glycerol.

**Figure 2 membranes-10-00307-f002:**
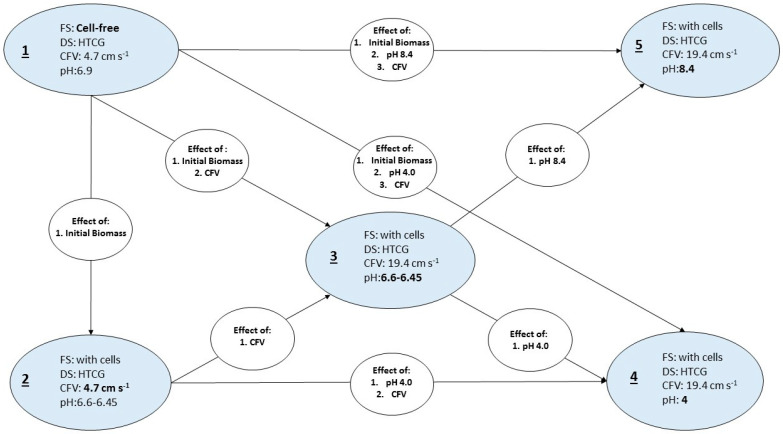
Graphical representation of the experimental setup and the possible interaction. FS: feed solution, DS: Draw solution, CFV: cross-flow velocity, HTCG: hexane-treated crude glycerol. Five setups were used with a one-variable-at-time experimental approach. The possible effects that can be studied are presented in the white oval shapes.

**Figure 3 membranes-10-00307-f003:**
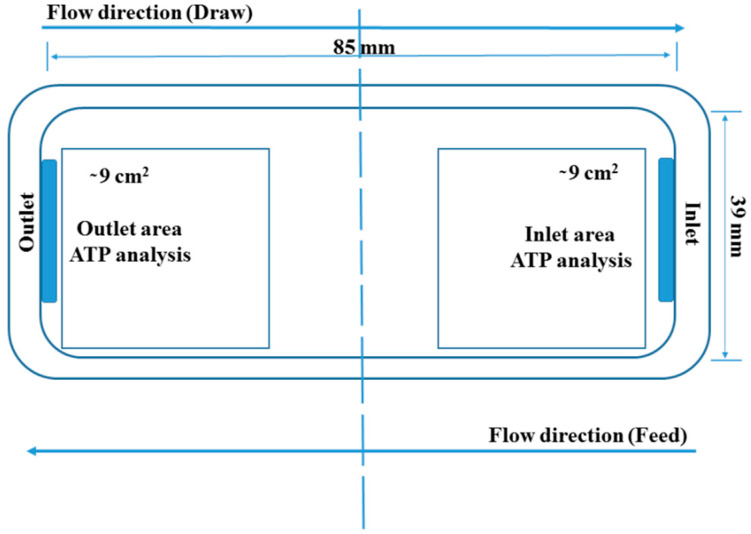
Graphical representation of the membrane area (active layer facing up). The dashed line indicates the area were the membranes were cut after filtration and the boxes the areas were ATP was extracted and quantified.

**Figure 4 membranes-10-00307-f004:**
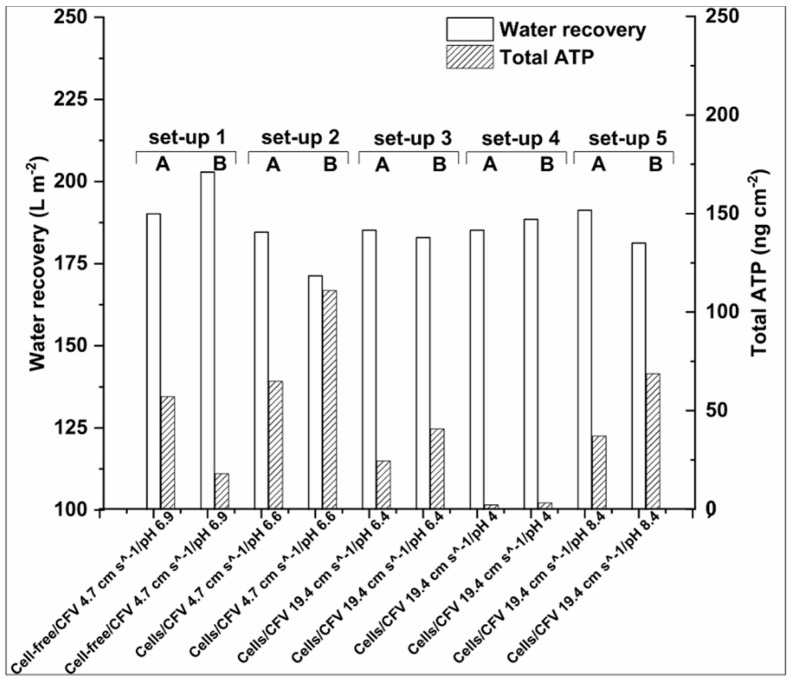
Water recoveries (white columns) and the corresponding Total-ATP concentration (shaded columns) after 22 h of FO operation. The letters A and B correspond to the replications of the experiments.

**Figure 5 membranes-10-00307-f005:**
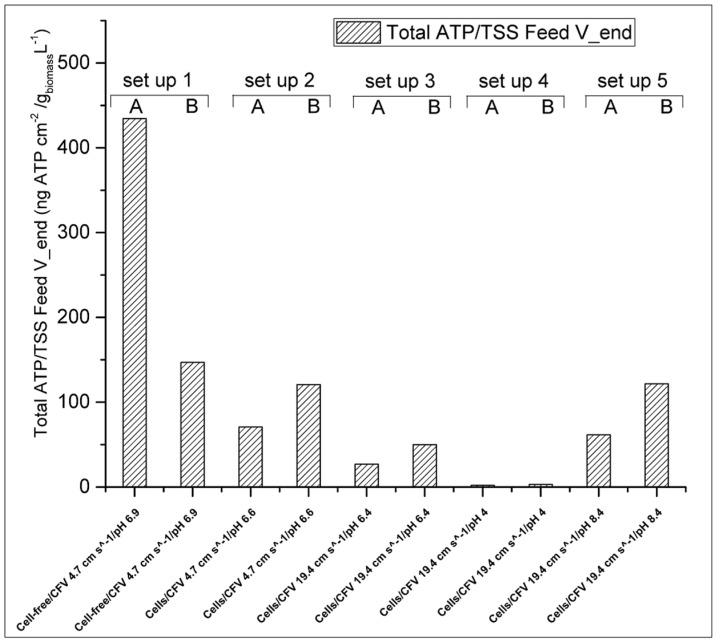
Total-ATP cm^−2^ normalized by the final TSS in the feed. The highest normalized Total-ATP concentration was found for set-up 1 followed by set-ups 2, 5, and 3. The lowest concentration of Total-ATP was found for set-up 4.

**Figure 6 membranes-10-00307-f006:**
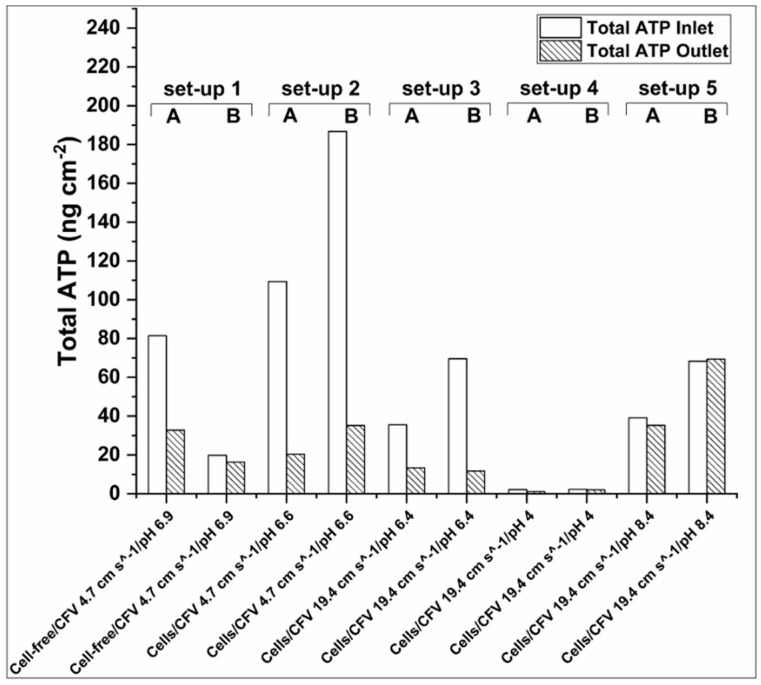
Total-ATP concentration on the inlet and outlet of the membrane patches. For set-ups 1, 2, and 3 there was a general tendency for higher Total-ATP concentration in the membrane inlet, whereas for set-up 5 Total-ATP concentration was similar in both ends of the membrane coupon. Very low amounts of Total-ATP were detected for set-up 4.

**Figure 7 membranes-10-00307-f007:**
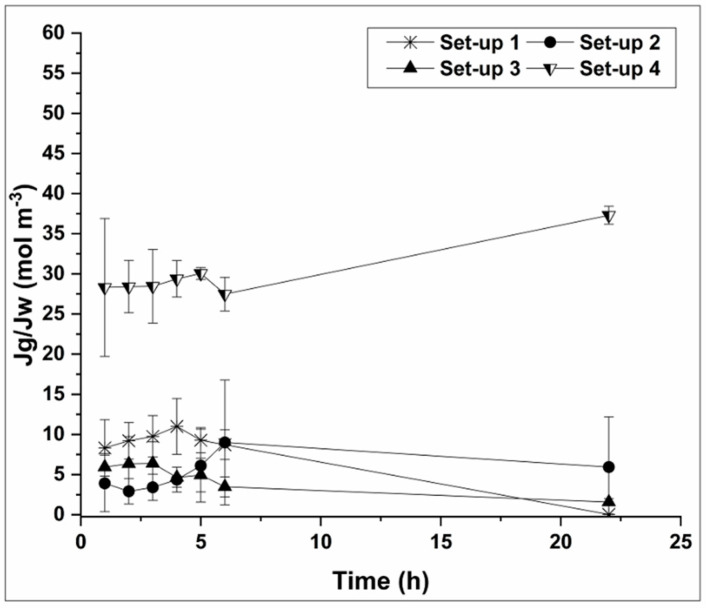
Specific reverse glycerol flux during different FO operation setups. The specific reverse glycerol flux for set-ups 1, 2, and 3 (high Total-ATP accumulation on the membrane coupons) was considerably lower than set-up 4 (very low Total-ATP accumulation on the membrane coupons). Specific reverse flux for set-up 5 was inconsistent between the replicates.

**Table 1 membranes-10-00307-t001:** Baseline experiments and membrane performance. The number in the “baseline set-up” column corresponds to the membrane coupon that subsequently used for the experimental setup with the conditions described in Figure 2 and the letter (A or B) the repetition of the experiment.

Baseline Setup	CFV (cm s^−1^)	*J_Waver_* (L m^−2^ h^−1^)	*J_Saver_* (g m^−2^ h^−1^)	*J_S_/J_W_* (g L^−1^)
1A	4.7	10.04	0.82	0.08
1B	4.7	9.81	0.87	0.09
2A	4.7	8.80	2.30	0.26
2B	4.7	10.61	1.82	0.17
3A	19.4	10.36	2.74	0.26
3B	19.4	10.38	2.64	0.25
4A	19.4	12.67	2.84	0.22
4B	19.4	10.95	2.67	0.24
5A	19.4	11.14	3.98	0.36
5B	19.4	10.61	2.92	0.28

**Table 2 membranes-10-00307-t002:** Initial and final pH of the corresponding feed solutions. A and B correspond to the repetitions of the experimental set-ups.

Setup	pH Feed_ini_	pH Feed_end_
1A	6.9	6.5
1B	6.9	6.5
2A	6.6	6.5
2B	6.45	6.45
3A	6.4	6.55
3B	6.4	6.5
4A	4.00	3.97
4B	4	4.1
5A	8.42	5.8
5B	8.50	6.10

**Table 3 membranes-10-00307-t003:** Intracellular vs. Total-ATP concentration on the membrane surface. ^1^ Total and Free-ATP was very low.

Setup	Intracellular ATP/Total ATP
1A	0.95
1B	0.79
2A	0.25
2B	0.42
3A	0.26
3B	0.47
4A	0.00 ^1^
4B	0.00 ^1^
5A	0.42
5B	0.42

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
