# Peer review of "Biofouling Mitigation Approaches during Water Recovery from Fermented Broth via Forward Osmosis"

_membranes, 2020, doi:10.3390/membranes10110307_

Round 1
Reviewer 1 Report
- The Introduction section needs to be revised more evidently to highlight the significance of this study over many prior studies. There is a lack of clarity for literature review on other existing papers, which will require to go through what they addressed and found thoroughly; what are the differences between them and yours. As far as I am concerned, many studies have identified the impact of hydrodynamic conditions on biofouling in osmotic driven processes.
- Sentences of Abstract in the manuscript are divided into three sections by marking (1) Background, (2) Methods, and (3) Conclusions. In my opinion, it seems not to be in the typical format of research journal articles, but that of academic conferences. Therefore, it would be better to remove those words and polish the writing a little.
- All references present in the manuscript needs to be rechecked carefully and used with more relevant papers. This is one example that I found: (Lines #111-113) References #1 and #18 are not closely related to biofouling mitigation approaches. Reference #18 does not deal with biofouling, but the organic and inorganic fouling control. Also, I could not find out any association between the authors' claim and Reference #1.
- Too obvious conclusions. Some conclusions are too general and commonsense that other previous studies on membrane fouling have already identified.
- Several spelling and mistakes still can be found in the manuscript. For instance, ‘set up’ is the verb as far as I know. In this case, either ‘setup’ or ‘set-up’ is correct as a noun form.
- For readability, all tables in the manuscript should be revised by framing it with a solid black line and the top rows in the tables.
Author Response
REVIEWER 1
Comment 1. The Introduction section needs to be revised more evidently to highlight the significance of this study over many prior studies. There is a lack of clarity for literature review on other existing papers, which will require to go through what they addressed and found thoroughly; what are the differences between them and yours. As far as I am concerned, many studies have identified the impact of hydrodynamic conditions on biofouling in osmotic driven processes.
Reply 1: The latter is true. To the authors’ opinion, the combination of the methods and experimental setup, together with the highly complex matrices applied, provides substantial novelty. A number of key relevant scientific publications are cited in the introduction with the objective to highlight some major aspects regarding biofouling in osmotic driven processes. By any means, it is not attempted to perform an extensive literature review of similar applications, since the current study is focused on a rather specific combination of draw and feed solutions, thus a direct comparison with prior literature cannot be conducted without severe limitations. Having this in mind and in order to comply with the reviewer comment, we have identified a recent relevant study and compared our results with those of Lee et al. (2020) in the results section of the revised manuscript. Moreover, we have added a couple of relevant publications in the literature section – see also our response to comment 3 of the same reviewer.
Comment 2. Sentences of Abstract in the manuscript are divided into three sections by marking (1) Background, (2) Methods, and (3) Conclusions. In my opinion, it seems not to be in the typical format of research journal articles, but that of academic conferences. Therefore, it would be better to remove those words and polish the writing a little.
Reply 2: The abstract had been drafted based on journal’s recommendations. In principle, we agree with the reviewer and therefore numbering and titles have been removed from the abstract. Also, minor writing polishing was applied to improve the flow of the text. Please see track changes.
Comment 3. All references present in the manuscript needs to be rechecked carefully and used with more relevant papers. This is one example that I found: (Lines #111-113) References #1 and #18 are not closely related to biofouling mitigation approaches. Reference #18 does not deal with biofouling, but the organic and inorganic fouling control. Also, I could not find out any association between the authors' claim and Reference #1.
Reply 3: References #1 & #18 had been cited as general references that introduce the reader in the overall landscape regarding osmotic pressure driven process and membrane fouling. Taking into account the reviewer’s comment, we have replaced references 1 and 18 in lines 111-120 with more specific ones, those of Kwan et al, 2015 and Wang et al, 2018 and Chun et al., 2017, respectively, in the revised version. We have also added the recent publication of Lee et al, 2020 in line 118. More relevant references are cited in the Results & Discussion section according to the topic that is addressed per subsection. In total 58 studies are cited, which is above the average for research papers. Still, the authors are open to suggestions if the reviewer can point to other specific studies than the ones already added in the revised version, which can contribute even more to discussions of results.
Comment 4. Too obvious conclusions. Some conclusions are too general and commonsense that other previous studies on membrane fouling have already identified.
Reply 4: Conclusion 1 was removed and merged with conclusion 3 (see Line 519). Also, conclusion 6 was removed since it is general and can be considered as commonsense.
Comment 5. Several spelling and mistakes still can be found in the manuscript. For instance, ‘set up’ is the verb as far as I know. In this case, either ‘setup’ or ‘set-up’ is correct as a noun form.
Reply 5: All the references in the text as ‘set up’ were corrected to ‘setup’. Also the text has been re-checked and corrections have been made. See track changes.
Comment 6. For readability, all tables in the manuscript should be revised by framing it with a solid black line and the top rows in the tables.
Reply 6: Tables 1, 2 & 3 were revised and a solid black line was added as a frame and on the top row

Reviewer 2 Report
- Lines 121-126: Since aquaporin-based TFC membranes are specially modified membranes that have usually higher water fluxes than normal TFC membranes, authors should describe why these special membranes were tested here, and whether they have a special influence / objective.
- Line 181 and Line 195: It would good if either Reynolds number is calculated or the channel height is added.
- Line 186-187: Why was the minimum JS/JW was set to 0.4 g.L-1? please add to the text.
- Section 1 and table: Please comment in the text on the relation between CFV and JSaver. Also, why were different CFV values applied here?
- Line 330: “at low CFV the initial deposition of foulants is more random than at high CFV”. What does “random” here mean? It may sound confusing. Higher fouling is always occurred at lower CFV and shear stress.
- Line 459: “… due to higher pressure drop at the beginning of membrane module” … this is not correct. At the leading module, the feed pressure (or applied pressure) is the highest, while the pressure drop increases for later modules.
Minor comments:
- Numbering and titles (e.g., background...) should be removed from the abstract.
- Tables formatting should be adjusted.
- Line 82: " because of... and due to ..." due to should be removed
- Line 204: “prior the” should be “prior to the”
- Line 331: Please correct “cross/flow”
Author Response
REVIEWER 2
Comment 1. Lines 121-126: Since aquaporin-based TFC membranes are specially modified membranes that have usually higher water fluxes than normal TFC membranes, authors should describe why these special membranes were tested here, and whether they have a special influence / objective.
Reply 1: Some of the benefits for using aquaporin-based TFC membranes are described in section 3.1 Baseline experiments, Lines: 271-277. Namely, their high water specificity due to the AqpZ channels, combined with the significant solute rejection and relatively high water flux, were considered to be key characteristics for their selection. Furthermore, the specific membranes have shown substantial glycerol rejection, during preliminary experiments, hence fulfilling the main objectives of the current study.
Comment 2. Line 181 and Line 195: It would good if either Reynolds number is calculated or the channel height is added.
Reply 2: Channel height has been added in line 172.
Comment 3. Line 186-187: Why was the minimum JS/JW was set to 0.4 g.L-1? please add to the text.
Reply 3: Preliminary experiments have shown that for a consistent performance, the minimum baseline reverse salt flux should not exceed 0.4 g.L-1. Also, and according to Blandin et al. 2016, as JS/JW 0.4 g.L-1 was within the performance range of other commercially available FO membranes. The justification is now added to the text.
Comment 4. Section 1 and table: Please comment in the text on the relation between CFV and JSaver. Also, why were different CFV values applied here?
Reply 4: A relevant sentence was added in lines 283-284. The different CFV we applied following the experimental set up shown at Figure 2. The reason for selecting those values, was to investigate the effect of a low CFV (represented by 4.7 cm s-1) and a high CFV (represented by 19.4 cm s-1) on membrane fouling.
Comment 5. Line 330: “at low CFV the initial deposition of foulants is more random than at high CFV”. What does “random” here mean? It may sound confusing. Higher fouling is always occurred at lower CFV and shear stress.
Reply 5: The use of the word “random” was chosen in an attempt to explain the more unpredictable deposition of biomass in low CFV, when compared to high CFV. Thus, the initiation of a biofouling layer development in short term experiments (i.e. 22h), expressed as total ATP accumulation, might not be as synchronized as it would have been if the filtration was conducted for several days. It was rephrased to “unpredictable” to improve clarity.
Comment 6. Line 459: “… due to higher pressure drop at the beginning of membrane module” … this is not correct. At the leading module, the feed pressure (or applied pressure) is the highest, while the pressure drop increases for later modules.
Reply 6.
We agree with the reviewer; what we meant was that the slope of the pressure drop profile is higher at the beginning of membrane module. This is now corrected in the text.
Minor comments:
Comment 1. Numbering and titles (e.g., background...) should be removed from the abstract.
Reply 1: All numbering and titles have been removed from the abstract. Please see track changes.
Comment 2. Tables formatting should be adjusted.
Reply 2: Tables 1, 2 & 3 were revised and a solid black line was added as a frame and on the top row
Comment 3. Line 82: " because of... and due to ..." due to should be removed
Reply 3: “due to” was removed from Line 84
Comment 4. Line 204: “prior the” should be “prior to the”
Reply 4: “to” was added
Comment 5. Line 331: Please correct “cross/flow”
Reply 5: “cross/flow” was corrected to “cross-flow”

Round 2
Reviewer 1 Report
The resubmitted manuscript has been appropriately revised with consideration for the raised comments.